# Enhancing Motor and Sensory Axon Regeneration after Peripheral Nerve Injury Using Bioluminescent Optogenetics

**DOI:** 10.3390/ijms232416084

**Published:** 2022-12-16

**Authors:** Anna Ecanow, Ken Berglund, Dario Carrasco, Robin Isaacson, Arthur W. English

**Affiliations:** 1Department of Cell Biology, Emory University School of Medicine, Atlanta, GA 30322, USA; 2Department of Neurosurgery, Emory University School of Medicine, Atlanta, GA 30322, USA

**Keywords:** axon regeneration, peripheral nerve injury, mice, retrograde labeling

## Abstract

Introduction—Recovery from peripheral nerve injuries is poor even though injured peripheral axons can regenerate. Novel therapeutic approaches are needed. The most successful preclinical experimental treatments have relied on increasing the activity of the regenerating axons, but the approaches taken are not applicable to many nerve-injured patients. Bioluminescent optogenetics (BL-OG) is a novel method of increasing the excitation of neurons that might be similar to that found with activity-dependent experimental therapies. We investigated the use of BL-OG as an approach to promoting axon regeneration following peripheral nerve injury. Methods—BL-OG uses luminopsins, light-sensing ion channels (opsins) fused with a light-emitting luciferase. When exposed to a luciferase substrate, such as coelenterazine (CTZ), luminopsins expressed in neurons generate bioluminescence and produce excitation through their opsin component. Adeno-associated viral vectors encoding either an excitatory luminopsin (eLMO3) or a mutated form (R115A) that can generate bioluminescence but not excite neurons were injected into mouse sciatic nerves. After retrograde transport and viral transduction, nerves were cut and repaired by simple end-to-end anastomosis, and mice were treated with a single dose of CTZ. Results—Four weeks after nerve injury, compound muscle action potentials (M waves) recorded in response to sciatic nerve stimulation were more than fourfold larger in mice expressing the excitatory luminopsin than in controls expressing the mutant luminopsin. The number of motor and sensory neurons retrogradely labeled from reinnervated muscles in mice expressing eLMO3 was significantly greater than the number in mice expressing the R115A luminopsin and not significantly different from those in intact mice. When viral injection was delayed so that luminopsin expression was induced after nerve injury, a clinically relevant scenario, evoked M waves recorded from reinnervated muscles were significantly larger after injury in eLMO3-expressing mice. Conclusions—Treatment of peripheral nerve injuries using BL-OG has significant potential to enhance axon regeneration and promote functional recovery.

## 1. Introduction

Peripheral nerve injuries (PNIs) are a prevalent clinical issue, impacting the quality of life of those who are injured and providing a significant economic burden for those affected [1]. Estimates suggest that some form of nerve injury, most commonly resulting from high-speed trauma, affects two million individuals in the United States alone (NIH Publication No. 18-NS-4853). Other age-associated etiologies include falls in the elderly and birth-related injuries in pediatric populations [1]. Despite the ability of axons in injured nerves to regenerate after PNIs, only ~10% of patients experience some degree of recovery. This leaves many individuals permanently disabled, with irreversible symptoms such as impaired motor function, loss of sensation, and often pain [2]. The extent to which functional recovery is not attained is often attributed to the fact that axon regeneration in injured nerves is slow and inefficient [3] and that this sluggishness is exacerbated if repair of the injured nerve is delayed [4]. Thus, novel treatments directed at enhancing axon regeneration are required to improve the outcomes for those with PNIs.

Based on the results of existing preclinical research, two experimental treatments, brief low-frequency (20 Hz) electrical stimulation [5,6] and moderate exercise [3,7,8,9], are especially effective in accelerating axon growth and improving functional recovery [10]. We [11,12] and others [5,13] have shown previously that increasing the activity of injured neurons is both necessary and sufficient to improve axon regeneration when using these experimental therapies, so they have come to be known as activity-dependent therapies [14]. Although these treatments are effective, there are barriers to their clinical use, such as delays prior to nerve repair surgery and co-morbidities that prevent their application [8]. Additionally, not all nerve-injured patients can be stimulated or enrolled in an exercise program. Thus, we have sought an alternative to these two successful activity-dependent experimental therapies.

Bioluminescent optogenetics (BL-OG) could be such an alternative. BL-OG uses luminopsins—fusion proteins of light-sensing ion channels (opsins) and light-emitting luciferase (Figure 1). The luminopsin used here is a red-shifted, highly sensitive channelrhodopsin variant from *Volvox carteri* (Figure 1: VChR1) fused with luciferase derived from *Gaussia princeps* (Figure 1: GLuc) through a short linker [15,16]. The resulting construct was slightly modified to include the trafficking signal from a neuronal potassium channel, for better membrane targeting, in front of an enhanced yellow fluorescent protein (Figure 1: EYFP) tag (enhanced LMO3 or eLMO3) [17]. When exposed to an appropriate luciferase substrate, such as coelenterazine (CTZ), bioluminescence is generated by the luciferase component and sensed by the opsin component, opening its pore and enabling an influx of cations. In this way, injured neurons induced to express an excitatory luminopsin could be activated directly to promote axon regeneration. Using transgenic mice engineered to express an excitatory luminopsin (eLMO3) exclusively in neurons, we showed previously [18] that when treated once with CTZ after sciatic nerve transection and repair, motoneurons were excited for as long as three hours and subsequent axon regeneration was enhanced. The axons of more motoneurons regenerated and successfully reinnervated target muscles in animals treated with a fully functioning luminopsin than in wild-type controls and those treated with a non-functioning (R115A) luminopsin in which the opsin component was mutated to remove its ability to respond to the bioluminescence generated by CTZ treatment [17,18]. The aim of this study was to investigate whether similar results could be achieved using BL-OG when using a viral vector to induce luminopsin expression either before or after PNI. We also evaluated whether BL-OG treatment would have a similar effect on the regeneration of muscle sensory axons. The successful application of BL-OG in this manner could move it closer to translation to the clinical treatment of PNI.

## 2. Results

### 2.1. BL-OG Enhances Regeneration of Motor Axons

The experimental protocol used in retrograde labeling experiments is shown diagrammatically in Figure 2. Following injection of the retrograde tracers CTB 555 into the gastrocnemius muscle (GAST) and CTB 647 into the tibialis anterior muscle (TA), motoneurons were identified in horizontal sections of the lumbar spinal cord containing these fluorescent markers (Figure 3A: red and blue cells, respectively), indicating that their motor axons had regenerated and successfully reinnervated those muscles. A very small number of motoneurons contained both retrograde tracers (Figure 3A: arrow), suggesting that their regenerating axons had branched and reinnervated both muscles. Motoneurons expressing the eLMO3 or R115A luminopsin constructs were also noted, marked by their immunoreactivity to the antibody against GFP (Figure 3A: green cells).

The significance of differences in the number of retrogradely labeled motoneurons from injections of tracers into both GAST and TA in the three groups studied (Intact, R115A, and eLMO3) was first evaluated using a one-way ANOVA. Significant differences were found for both motoneuron groups (GAST, F_2,13_ = 7.392, *p* < 0.01; TA, F_2,13_ = 25.17, *p* < 0.0001). Using post hoc paired testing (Tukey), the number of retrogradely labeled motoneurons in mice injected with the vector encoding the functioning eLMO3 and treated with CTZ was significantly greater than the number found in similarly treated mice that had been injected with the R115A mutant luminopsin construct, for both GAST and TA (Figure 3B). Additionally, for both muscles, there was no significant difference between the number of successfully regenerating motoneurons from mice exposed to the viral vector encoding eLMO3 and those from intact controls. This indicates not only that the BL-OG treatment had a positive effect on motor axon regeneration but suggests near complete muscle reinnervation by four weeks after injury when treated with BL-OG. For the small number of cells which contained both retrograde tracers, there was no significant difference between groups (ANOVA, F_2,13_ = 0.574, *p =* 0.5772), suggesting that BL-OG treatment did not affect this double reinnervation.

Because the motoneuron soma size measurements were not normally distributed (Figure 3C), a non-parametric test (Kruskal–Wallis ANOVA) was performed to determine whether any significant differences between groups were present. There were no significant size differences between groups in motoneurons reinnervating GAST, but for motoneurons innervating the TA a significant difference was found (H(3) = 8.807, *p <* 0.012). Based on post hoc testing (Dunn’s), the sizes of labeled motoneurons were significantly smaller in mice exposed to the R115A luminopsin construct than in either intact control mice or mice injected with the eLMO3-expressing vector. Fewer of the largest TA motoneurons regenerated axons effectively in the mice expressing R115A.

Although more motoneurons were retrogradely labeled in the BL-OG-treated mice, the total number of motoneurons immunoreactive for GFP in sections of spinal cords was very small for both eLMO3 and R115A mice (Figure 3D). The proportion of retrogradely labeled motoneurons that contained this marker of luminopsin expression (11.84% ± 6.59% SEM) was also smaller than might be anticipated to explain the substantial differences noted above. Thus, the exact mechanism of BL-OG enhancement of motor axon regeneration remains unclear.

### 2.2. BL-OG Facilitates Recovery of Neuromuscular Function

Compound muscle action potentials (direct muscle responses or M waves) were evoked in the GAST and TA muscles by sciatic nerve stimulation. In mice that had been induced to express eLMO3 and treated once with CTZ, M waves recorded four weeks after transection and repair of the sciatic nerve were much larger than those recorded at the same time from mice induced to express the mutant R115A luminopsin (Figure 4A). Maximum amplitudes of full-wave rectified M waves (MMAX) recorded at four weeks after injury were scaled to the same measures recorded prior to sciatic nerve injury. These scaled M response amplitudes were compared between the eLMO3- and R115A-expressing mice. Based on the results of a one-way ANOVA (F_3,16_ = 3.448, *p* < 0.0418) and post hoc paired (Tukey) testing, scaled M-wave amplitudes were significantly larger in mice induced to express eLMO3 than in the animals induced to express R115A (Figure 4B). Amplitudes of M waves in the R115A mice had recovered to ca. 16% of pre-injury amplitudes, a value comparable to those reported for untreated animals at four weeks post-injury [11,19]. In eLMO3-expressing mice, however, M-wave amplitudes were greater-than-fourfold larger in this measure of functional recovery compared with the R115A group.

### 2.3. BL-OG Treatment Enhances Regeneration of Sensory Axons

We counted sensory neurons labeled with the fluorescent retrograde tracers in histological sections of the L4 and L5 dorsal root ganglia (DRG) (Figure 5A). Using a one-way ANOVA, significant differences in the number of retrogradely labeled neurons innervating GAST (F_2,13_ = 16.59, *p* < 0.0003) and TA (F_2,13_ = 17.85, *p* < 0.0002) (Figure 5B) were found. Based on post hoc paired testing (Tukey), axons of significantly more DRG neurons had regenerated successfully in animals injected with the virus encoding eLMO3 than in those injected with the virus encoding the mutated R115A luminopsin (Figure 5B). We found no significant difference in the number of retrogradely labeled sensory neurons between the eLMO3-expressing mice and intact controls, suggesting a complete sensory reinnervation in those animals. A very small number of neurons contained both retrograde tracers, with no significant differences between groups.

In contrast to the observation for motoneurons, above, GFP immunoreactivity, indicating the expression of R115A or eLMO3, was found in a large proportion of DRG neurons. Among retrogradely labeled neurons, as many as half also expressed immunoreactivity to GFP (Figure 5C).

We measured the soma cross-sectional areas of all retrogradely labeled DRG neurons. We did not presume that this measure of neuron size would enable us to distinguish between neurons of different functional classes [19,20], but we used it simply as a means of investigating any bias of the successful BL-OG treatment toward neurons of different sizes. The distributions of sizes of labeled DRG neurons are shown in Figure 5D. Because these cell sizes were not normally distributed, a non-parametric (Kruskal–Wallis) ANOVA was used to compare medians between groups. No significant differences were found in neurons labeled from TA (H(3) = 2.995, *p* = 0.2282), but for DRG neurons labeled from GAST, a significant difference was found (H(3) = 9.256, *p* < 0.0085). Using post hoc paired testing (Dunn), we found that the median size of GAST DRG neurons in mice induced to express eLMO3 was significantly smaller than that of mice expressing the mutant R115A luminopsin, but not that of intact controls. In comparing DRG soma sizes between GAST and TA, median sizes were found to be significantly larger in DRG neurons labeled from GAST (Figure 5D).To investigate size further, we determined the proportion of all labeled DRG cells in three different size groupings [21]: small (<300 µm^2^), medium-sized (300–700 µm^2^), and large (>700 µm^2^) (Figure 5E). Within each size grouping, no significant differences were found between intact, R115A, and eLMO3 animals for either GAST or TA. However, in comparing proportions between GAST and TA, significant differences were found (F_5,12_ = 60.66, *p* < 0.0001). Proportionally, more neurons in the small size grouping were labeled from TA whereas intermediate- and large-sized neurons were preferentially labeled by GAST.

### 2.4. Post-Injury Induction of BL-OG Promotes Recovery of Neuromuscular Function

The results presented above are all based on experiments in which an excitatory luminopsin, or a mutated version of it, was expressed in neurons with axons in a peripheral nerve prior to the injury of that nerve. Using a delayed reinnervation model, we evaluated the use of BL-OG when luminopsin expression was induced after the nerve injury. One of the main branches of the sciatic nerve, the common fibular nerve, was cut and ligated, and then a viral vector expressing either eLMO3 or R115A was injected into its proximal stump. A control group of mice received no viral injection. After allowing four weeks for retrograde transport and viral transduction, the other main branch of the sciatic nerve was cut and the freshly trimmed proximal stump of the common fibular nerve was aligned to its distal stump (Figure 6A). At that time, mice were treated with CTZ or left untreated. Four weeks later, the extent of motor reinnervation of the GAST muscle was assayed using the amplitudes of M waves evoked by stimulation of the sciatic nerve proximal to injury. Examples of M waves recorded from these animals are shown in Figure 6B. Mean (± SEM) maximum M-response amplitudes are shown for the different treatment groups in Figure 6C. Based on the results of a one-way ANOVA (F_4,19_ = 4.057, *p* < 0.0153) and post hoc paired (Tukey) testing, M-wave amplitudes were significantly larger in the mice induced to express eLMO3 after injury and also receiving CTZ than each of the other groups. No significant differences were found between the other groups. Thus, BL-OG treatment resulted in enhanced recovery of neuromuscular function even when it was initiated after nerve injury.

## 3. Discussion

Poor recovery from peripheral nerve injuries remains a significant clinical problem, leaving many individuals with permanent symptoms and disabilities [2]. As such, the development of novel treatments aimed at enhancing axon regeneration could improve outcomes for those affected [3]. Increasing the activity of injured neurons is effective in promoting this enhancement [11,12]. Because some barriers exist to the clinical translation of experimental activity-dependent therapies, such as exercise or low-frequency electrical stimulation, we investigated BL-OG as an alternative to improve axon regeneration by directly stimulating injured neurons. In this study, we aimed to investigate the feasibility of enabling BL-OG via viral vector injection of a luminopsin construct.

Experimental treatments to enhance axon regeneration after PNI are considered encouraging if they promote the effective regeneration of axons of more injured neurons leading to improved functional outcomes. One of the main findings of this study was an increased number of both motor and sensory neurons whose axons successfully regenerated and reinnervated their muscle targets in animals expressing the fully functioning eLMO3, compared to those induced to express the mutated R115A luminopsin. The increased number of motoneurons seen here was consistent with the results of our previous study investigating the use of BL-OG in transgenic mice [18]. It is also consistent with the results of previous studies evaluating activity-dependent experimental therapies such as electrical stimulation [13] or moderate exercise [22]. However, this is the first study that investigates the effectiveness of BL-OG treatments using the viral induction of luminopsin in wild type mice, an important consideration for any future translation of BL-OG to clinical use. No significant difference in the number of sensory and motor neurons innervating two muscle targets exists between the injured and intact sides of the mice treated with BL-OG. This suggests a near complete reinnervation by *both* sensory and motor neurons only four weeks after nerve injury in mice treated with BL-OG. An additional important finding is that BL-OG treatment in this manner resulted in a greater-than-four-fold restoration of neuromuscular function four weeks after injury than controls. Full muscle reinnervation after sciatic nerve injury in mice, based either upon retrograde labeling [22] or restoration of M-wave amplitudes [8,18], takes considerably longer without treatment. It is now well established that at least some of the slowness of axon regeneration after PNI is due to a delay of days or weeks before some regenerating axons enter the distal nerve segment [5]. The clear effect of BL-OG treatment in reducing this “temporal staggering” of regenerating motor [5,23] and sensory [24] axons after PNI is remarkably similar to that demonstrated for the use of low-frequency electrical stimulation in rats. Based on these results, we consider BL-OG a promising potential therapy for enhancing axon regeneration after PNI.

Because there was no significant difference between median sizes of successfully regenerating DRG neurons in BL-OG-treated and intact mice, this treatment seemingly does not alter the success of axon regeneration of sensory neurons of different sizes in either GAST or TA. By comparing the distribution of sensory neurons across three size groupings (<300 μm^2^, 300–700 μm^2^, and >700 μm^2^) [21], no significant differences were noted between the animals injected with the virus encoding the excitatory luminopsin and those injected with the virus encoding the R115A luminopsin. Significant differences were found in the proportions of DRG neurons in the different size classes between TA and GAST, but not between treatment groups. To the extent possible based only on size analysis, treatment with BL-OG after PNI restores the sensory neuron population to these two muscles appropriately.

All of these studies were performed in mice in which luminopsin expression was induced prior to nerve injury. While these results support the feasibility of BL-OG to treat PNI, any proposed clinical application of BL-OG to treat PNI will require that luminopsin expression be induced after the injury. The results of our experiments, using delayed cross-reinnervation, addressed this concern. The induction of BL-OG by CTZ treatment, even after a four-week delay in the repair of a cut nerve to allow for post-injury induction of luminopsin expression, resulted in a 3.5-fold increase in M-wave amplitude, a commonly used clinical measure of neuromuscular function, relative to controls. While these results are very encouraging, they also must be considered as preliminary. More extensive study of dose and dosing of CTZ will be required in a clinically relevant model such as the one used here, but the current findings offer important evidence to support the feasibility of using BL-OG as a treatment for PNI.

While the increased number of successfully regenerating axons of motoneurons observed here after BL-OG treatment is promising, we were unable to attribute this positive effect specifically to luminopsin expression. The proportion of retrogradely labeled motoneurons expressing a luminopsin construct, as defined by immunoreactivity to GFP, was smaller than could account for the extent of improvement in motor axon regeneration produced by the BL-OG treatment. This lack of correlation between the extent of EYFP-expressing motoneurons and the increase in the numbers of retrogradely labeled motoneurons in BL-OG-treated mice significantly limits any mechanistic explanations of the enhancement of axon regeneration observed. It is possible that the level of luminopsin expression in many motoneurons was small and we were simply unable to detect the EYFP tag on it. Our difficulty in visualizing the luminopsin would have been further complicated if it became distributed throughout the membranes of the extensive dendritic arbors found in motoneurons. Future studies using methods such as PCR or further amplifying the GFP signal will need to be employed to improve the ability to detect the virally induced luminopsin in motoneurons. However, it is also possible that even a relatively weak luminopsin expression might be sufficient to promote motor axon regeneration when excited after CTZ administration.

The extent of luminopsin expression was much greater in dorsal root ganglion cells, which do not have the elaborate dendritic arbors found in motoneurons. Nearly half of the retrogradely labeled DRG neurons studied also expressed GFP immunoreactivity, a proportion that could be sufficient to account for the increase in sensory axon regeneration after BL-OG treatment. Given the disparity between the demonstrable identification of expression of the luminopsins in sensory and motor neurons, one might speculate that the effectiveness of the BL-OG treatment on sensory and motor axon regeneration could be due to different cellular mechanisms. Enhanced sensory axon regeneration could be the result of direct excitation of DRG neurons expressing eLMO3. Enhanced motor axon regeneration could be due to a combination of a direct excitation of those motoneurons expressing eLMO3 and an indirect excitation of motoneurons by eLMO3 expressing sensory neurons. The combination of direct excitation in motoneurons faintly expressing luminopsin and the excitation produced by CTZ-driven activity in DRG neurons projecting to them might be adequate to promote motor axon regeneration. At the time of administration of CTZ in our experiments, such excitatory connections continue to be functional [25]. Although we showed previously, using optogenetics, that the activation of DRG neurons in the absence of increased motoneuron excitation was not sufficient to promote motor axon regeneration after PNI [11], inducing excitation of motoneurons that might already be weakly excited, via direct connections from DRG neurons, remains a possibility.

In conclusion, we aimed to investigate whether, when inducing expression of an excitatory luminopsin either before or after PNI, treatment using BL-OG would promote subsequent axon regeneration. We report here that BL-OG treatment in such circumstances did significantly enhance the regeneration of the axons of motor and sensory neurons after peripheral nerve injury. We believe that the results presented here have moved the use of the therapy closer towards potential clinical applications, and the inclusion of sensory neurons and regenerating cell soma sizes provided valuable new information on the action of BL-OG. The use of BL-OG in enhancing regeneration after peripheral nerve injury remains a promising avenue through which those affected could regain function.

## 4. Materials and Methods

### 4.1. Animals and Surgeries

Eight intact C57B6/J mice (four female, four male) ranging from 6–13 weeks of age were used in the retrograde labeling experiments. Mice were anesthetized using isoflurane, and their right sciatic nerves were exposed in the mid-thigh. The exposed nerves were injected above the branching of the tibial and common fibular nerves, with 1–2 µL of an adeno-associated (AAV2/9) viral vector encoding either an excitatory luminopsin (eLMO3) (1.2 × 10^14^ vg/mL) (four mice) or a luminopsin with a mutated opsin component (R115A) (3.5 × 10^14^ vg/mL) [18] (four mice). Both constructs were under the control of the human synapsin (Hsyn) promoter and were derived from *Volvox* channelrhodopsin 1 (VChR1) fused with *Gaussia* luciferase (GLuc) through a short linker [15,17] in front of the enhanced yellow fluorescent protein (EYFP) tag (Figure 1).

The mutated luminopsin sequence is identical to that of the eLMO3 construct except for a single amino acid perturbation (arginine to alanine at location 115, R115A) in the VChR1 component. In this mutant, the luciferase component of the luminopsin generates bioluminescence in the presence of CTZ but the channelrhodopsin component does not respond to that light or activate neurons [17,18]. The use of this R115A luminopsin allowed us to ensure that any effect could be attributed to neuronal excitation, rather than other components of the BL-OG system [18]. The use of this mutant also acts as a control for any effect of CTZ alone. CTZ has been shown to have antioxidant properties, so its presence might affect regeneration [26]. Even though CTZ treatment alone did not enhance axon regeneration in transgenic mice [18], it is important to control for this possibility.

After waiting two weeks for retrograde viral transport and neuronal transduction, injected sciatic nerves were cut in the mid-thigh and repaired by simple end-to-end anastomosis, as we have described in more detail elsewhere [27]. The contralateral sciatic nerve in each mouse was not injected or injured and served as an intact control. Immediately after the repair surgery was completed, the mouse was administered a single dose of CTZ (10 mg/Kg, i.p.). The CTZ used was Inject-A-Lume (NanoLight Technologies), a form of native coelenterazine formulated for in vivo applications. In an earlier study [18], we showed that an injection of this dose of CTZ resulted in a rapid and long-lasting increase in bioluminescence over the spinal cord and sciatic nerve of transgenic mice expressing the same luminopsin constructs. It did not increase spontaneous neuromuscular activity in these mice or in mice induced to express the luminopsin using the same viral vectors employed here, which would be associated with increased motoneuron firing, but it did lower the threshold for reflex excitation of luminopsin-expressing motoneurons, suggesting an increase in their excitability [18]. This change in excitability reached a peak 45 min after CTZ injection and returned to baseline in three hours [18]. Because of these previously published findings, we assumed that the CTZ treatments employed in the present study would result in a similar increase in neuronal excitability.

Four weeks after nerve repair and CTZ treatment, different retrograde fluorescent tracers were injected into the two heads of the gastrocnemius and the tibialis anterior muscles to mark motor and sensory neurons whose axons had reinnervated these muscles. This survival time was chosen to be compatible with our previous studies [28,29,30] and others [5,8,14] on activity-dependent experimental therapies to enhance peripheral axon regeneration after peripheral nerve injury. Two microliters of a 1% solution of the beta subunit of cholera toxin, conjugated to Alexa Fluor 555 (CTB 555), was injected into the gastrocnemius muscle (GAST) (4 µL total), and 2 µL of a similar reagent, but conjugated to Alexa Fluor 647 (CTB 647), was injected into the tibialis anterior muscle (TA). Injections were made at several sites in each muscle and the injection needle was left in place for five minutes at each site to prevent seepage of tracer out of the muscle along the needle track. Similar injections into the contralateral muscles served to mark motor and sensory innervation from intact sources. After the last injection, each surgical site was washed three times with normal saline and surgical wounds were closed in layers before animals were returned to their cages. Mice were euthanized using an intraperitoneal injection of Euthasol (pentobarbital sodium and phenytoin sodium, 150 mg/Kg) perfused with 4% paraformaldehyde solution five days after retrograde tracer injections. The entire sequence of experiments is shown diagrammatically in Figure 2.

### 4.2. Tissue Processing and Immunohistochemistry

Following euthanasia, entire lumbar spinal cords, as well as right and left L4 and L5 DRGs, were collected from each animal and preserved in 20% sucrose solution. Spinal cords and DRGs were serially sectioned on a cryostat at 50 μm and 30 μm thickness, respectively, and mounted onto glass microscope slides. All sections were saved and used in analyses. Spinal cord tissue was reacted with antibodies to GFP to amplify visualization of the EYFP marker on the eLMO3 or R115A constructs. Slides were incubated with blocking solution containing 0.3% Triton X-100 and 10% normal goat serum for one hour before being incubated with primary antibody (Invitrogen, anti-GFP rabbit IgG, #A-11122, diluted 1:500 in phosphate-buffered solution (PBS), pH 7.4) at 4 °C overnight. Tissue was then washed three times with PBS before being incubated with secondary antibody (Invitrogen Goat anti-Rabbit 488, A-11008, 1:300) for 1 h, before being washed three more times.

### 4.3. Imaging

Images of sections of spinal cords were captured using a Leica DM6000 microscope, a Hamamatsu ORCA camera, and HCImage software. Images of DRG sections were captured using a Keyence BZ-X microscope. Motoneurons that were retrogradely labeled, indicating that their axons had successfully regenerated and reinnervated the gastrocnemius or tibialis anterior muscle, were counted and their sizes measured. Motoneurons were scored as retrogradely labeled if the fluorescent label filled the cell body and extended into the proximal dendrites, and contained a visible nuclear shadow (Figure 3A) [22]. A note was made of any neurons that contained both GFP (indicating presence of the luminopsin) and a retrograde label. Counts of DRG neurons, using the same criteria for inclusion, were conducted as well to determine the effect that the BL-OG treatment had on the regeneration of muscle sensory axons. Both motor and sensory neuron soma sizes were measured using Fiji software to determine whether the successfully regenerating neurons differed in size from those found on the intact sides of the animals. Images of adjacent microscope fields were stitched together using the Fiji plugin [31].

### 4.4. Electrophysiology

To evaluate the extent of restoration of neuromuscular function after PNI, compound muscle action potentials (direct muscle responses or M waves) were recorded from the GAST and TA muscles in response to stimulation of the sciatic nerve. The methods used have been described in more detail elsewhere [32]. In isoflurane-anesthetized mice, sciatic nerves were surgically exposed in the mid-thigh and paired needle electrodes (Ambu #74325-36/40, Columbia, MD, United States) were placed in contact with the nerve. Bipolar fine-wire EMG electrodes [33] were inserted through the skin into the lateral gastrocnemius and tibialis anterior muscles. Ongoing activity recorded from these muscles was sampled at 10 KHz using a laboratory computer system running custom Labview software and, when activity over a 10 ms period was within a user-defined background range, the computer delivered a single brief (0.3 ms) constant voltage pulse to the nerve via the needle electrodes and recorded EMG activity for 50 ms. A range of stimulus intensities was applied, from subthreshold to supramaximal. To avoid fatigue, stimuli were delivered no more frequently than once every five seconds. Amplitudes of M waves were measured as the average full-wave rectified voltage between the onset and duration of the recorded triphasic action potential.

Two sets of experiments used M waves as outcome measures. In one set, M waves were recorded from GAST and TA in 10 intact mice and then viral vectors encoding either eLMO3 or R115A were injected into their sciatic nerves (five mice each), as described above for the experiments using retrograde labeling. Four weeks later, the sciatic nerves of these mice were cut and repaired, as described above, and the mice were treated once with CTZ (10 mg/Kg, i.p.). After four more weeks, M waves were recorded from the reinnervated GAST and TA muscles. The amplitude of the largest M wave (M_MAX_) recorded from a reinnervated muscle was expressed as a proportion of the M_MAX_ recorded from that muscle in that mouse prior to injury. These scaled M_MAX_ values from mice induced to express eLMO3 and mice induced to express R115A were then compared using a one-way ANOVA, with post hoc paired (Tukey) testing.

In a second set of experiments, M waves were used to evaluate the restoration of neuromuscular function when the induction of luminopsin expression was performed after the nerve injury. An outline of these experiments is shown in Figure 6A. In isoflurane-anesthetized mice, the common fibular nerve was cut and ligated. Viral vectors encoding either eLMO3 or R115A (five mice each) were injected unilaterally into the proximal segment of the cut nerve near its bifurcation from the tibial nerve. In an additional five mice, nerve transection and ligation was performed but no viral injection was made. After allowing four weeks for retrograde transport and viral transduction, the injury site was exposed in isoflurane-anesthetized animals and the tibial nerve was transected. The ligated portion of the common fibular nerve was then trimmed with sharp scissors to remove the ligation and was aligned with the distal segment of the cut tibial nerve. The cross-repaired nerve was then secured in place using fibrin glue [27]. Animals that had received virus injections were then either administered a single dose of CTZ (10 mg/Kg, i.p.) or left untreated. Animals that did not receive a viral injection remained untreated. Four weeks later, M waves were recorded from the reinnervated GAST muscles of all the mice. Comparisons of the amplitudes of M_MAX_ between mice in different groups were made using a one-way ANOVA, with post hoc paired (Tukey) testing.

### 4.5. Statistical Methods

Numbers of animals in all experimental groupings used were deemed adequate based on a post hoc power analysis performed using G * Power (Power = (1-β err prob) > 0.8). GraphPad Prism software was used for statistical analyses. Significance was set as *p* < 0.05 in all analyses.

## Figures and Tables

**Figure 1 ijms-23-16084-f001:**
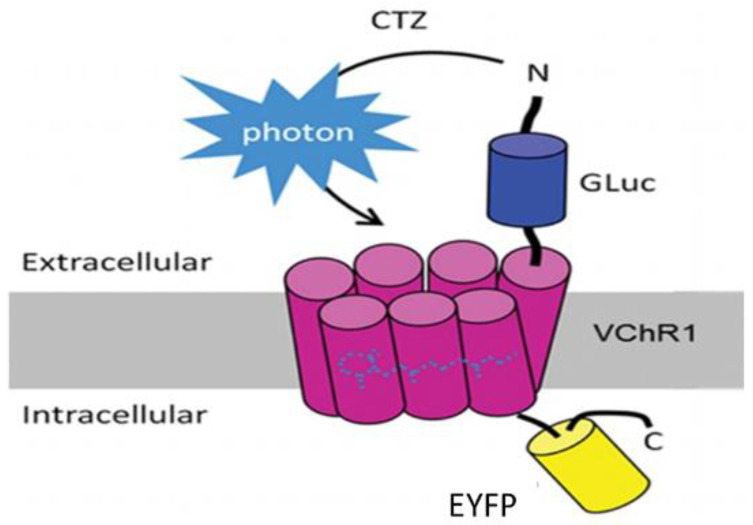
Diagram of the excitatory luminopsin (eLMO3) after [17]. Gaussia luciferase (GLuc) is fused to Volvox light-sensitive channel rhodopsin (VChR1). A fluorescent marker (EYFP) is added to the cytosolic terminal for visualization of expression. When exposed to a luciferase substrate, such as coelenterazine (CTZ), bioluminescence is produced and the cation channel of VChR1 is opened, exciting the neuron.

**Figure 2 ijms-23-16084-f002:**
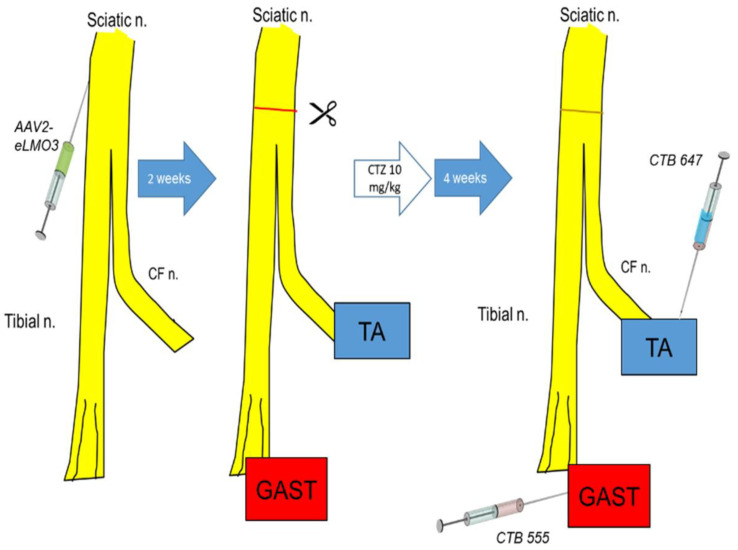
Diagram of the protocol used in the retrograde labeling experiments. An adeno-associated viral vector encoding either an excitatory luminopsin (eLMO3) or a non-functional mutant (R115A) was injected into the sciatic nerve. Two weeks later, the sciatic nerve was cut and repaired, and animals were treated with coelenterazine (CTZ). Four weeks later, fluorescent retrograde tracers were injected into the reinnervated tibialis anterior (TA) and gastrocnemius (GAST) muscles to mark sensory and motor neurons whose axons had regenerated and successfully reinnervated those targets.

**Figure 3 ijms-23-16084-f003:**
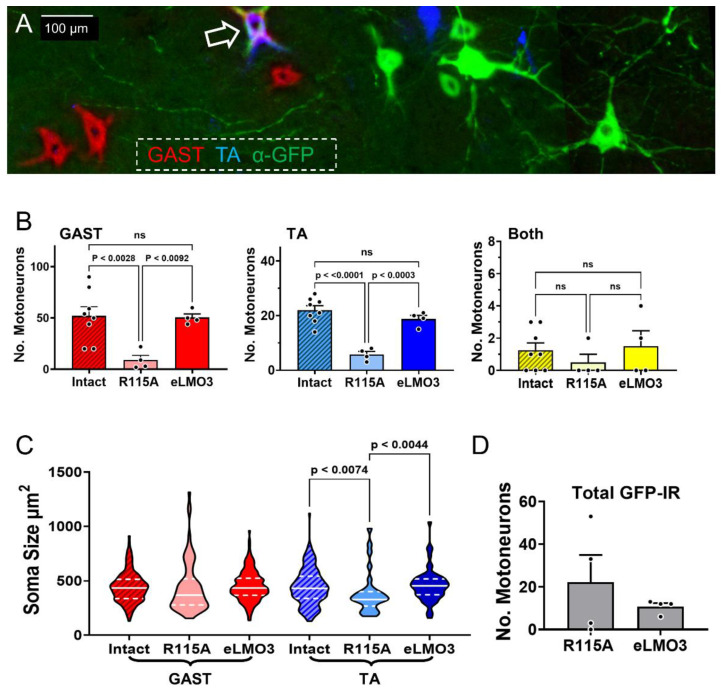
BL-OG treatment enhances the regeneration of motor axons. (**A**) An image of a longitudinal section of one side of the spinal cord of a mouse six weeks after injection of the eLMO3 construct into its sciatic nerve and four weeks after nerve transection and repair. Retrogradely labeled motoneurons are shown by the red (GAST) and blue (TA) fluorescent cells. Immunoreactivity to GFP (α-GFP), indicative of eLMO3 expression, is present in the green cells. One cell in the image (arrow) contains both retrograde labels and α-GFP. (**B**) Mean (+SEM) counts of retrogradely labeled motoneurons innervating the GAST (**left**), TA (**center**), and containing both tracers (**right**) in sections of spinal cords from intact mice, and reinnervated mice expressing either eLMO3 or the mutant R115A luminopsin. Significance of differences in means was evaluated using a one-way ANOVA, with post hoc paired (Tukey) testing where appropriate. (**C**) Distributions of soma sizes of labeled motoneurons in the three treatment groups are shown for GAST (**left**) and TA (**right**). The solid white line through each violin in the graph marks the median of that distribution. Dashed lines are located at quartiles. (**D**) The total numbers of motoneurons identified as expressing GFP immunoreactivity are shown for the mice induced to express R115A or eLMO3.

**Figure 4 ijms-23-16084-f004:**
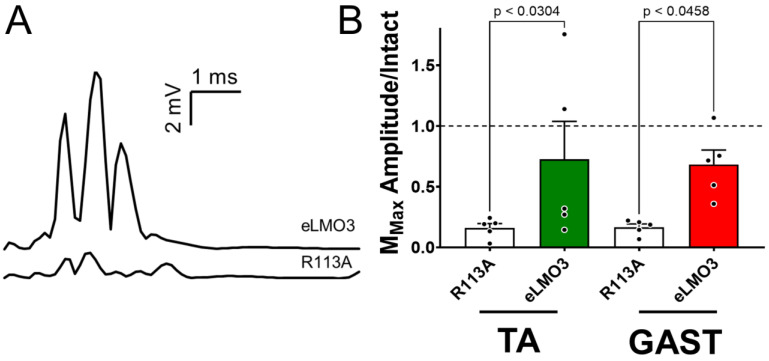
(**A**) Examples of compound muscle action potentials (M waves) recorded from the TA and GAST muscles four weeks after transection and repair of the sciatic nerve and treatment with coelenterazine. Sciatic nerves of mice had been injected two weeks prior to injury with Hsyn-AAV, encoding either an excitatory luminopsin (eLMO3) or a non-functional mutated version (R115A). (**B**) M-wave amplitudes were scaled in each mouse to the amplitude of the M wave recorded prior to injury. Mean (+SEM) scaled amplitudes for the two muscles and two treatments are shown.

**Figure 5 ijms-23-16084-f005:**
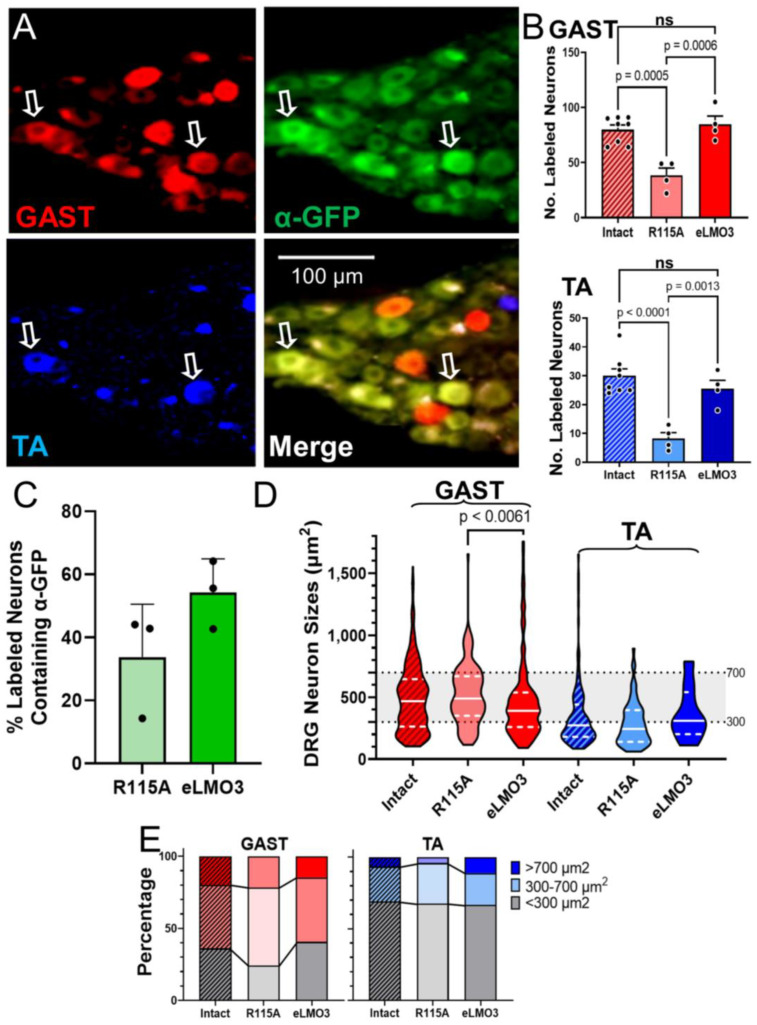
BL-OG treatment enhances the regeneration of muscle sensory axons. (**A**) A montage of an image of a section through the L4 DRG of a mouse six weeks after the injection of the eLMO3 construct into its sciatic nerve and four weeks after nerve transection and repair and treatment with CTZ. Retrogradely labeled sensory neurons are shown by the red (GAST) and blue (TA) fluorescent cells. Immunoreactivity to GFP (α-GFP), indicative of eLMO3 expression, is present in the green cells. Two cells (arrows) contain both retrograde labels and α-GFP. (**B**) Mean (+SEM) counts of retrogradely labeled sensory neurons innervating GAST (**top**) and TA (**bottom**) in L4 and L5 DRGs from intact mice, and reinnervated mice expressing either the mutant luminopsin (R115A) or eLMO3. (**C**) The mean (+SEM) percentage of L4 and L5 DRG neurons that contained a retrograde tracer and expresses GFP immunoreactivity are shown for mice induced to express R115A or eLMO3. (**D**) Distributions of soma sizes of retrogradely labeled sensory neurons in the three treatment groups are shown for GAST (**left**) and TA (**right**). The solid white line through each violin in the graph marks the median for that distribution. Dashed white lines are located at quartiles. The shaded area in the background delineates the region of intermediate sized (300–700 µm^2^) ganglion cells. (**E**) The proportions of DRG neurons retrogradely labeled from GAST (**left**) and TA (**right**)in different size classes are shown for the three different groups.

**Figure 6 ijms-23-16084-f006:**
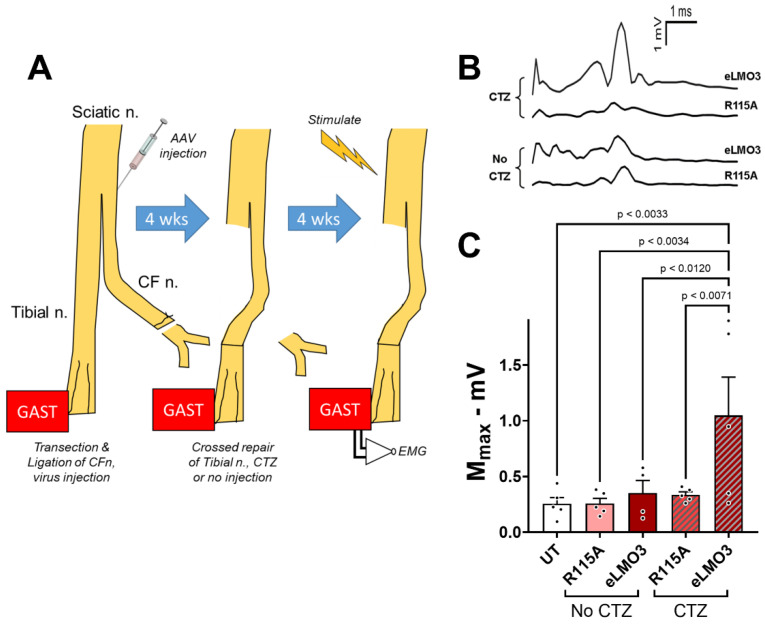
(**A**) Delayed nerve repair experiment. The common fibular nerve (CFn) was cut and ligated and an Hsyn-AAV encoding either LMO3 or the non-functional mutant (R115A) was then injected into the proximal stump of the cut nerve. A control group received no virus injection. Four weeks later, the un-ligated proximal stump of the CFn was attached to the distal segment of a freshly cut tibial nerve. Animals were then treated either with coelenterazine or left untreated. Four weeks later, compound muscle action potentials (M waves) were elicited in the GAST muscle by electrical stimulation of the sciatic nerve proximal to the injury site. (**B**) Examples of M waves recorded from the GAST muscles four weeks after delayed repair of the sciatic nerve. (**C**) Mean (+SEM) amplitudes from the untreated mice and from mice in the four experimental groups are shown.

## Data Availability

The data presented in this study are openly available in FigShare, https://doi.org/10.6084/m9.figshare.20597478 (accessed on 15 December 2020).

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
