# Peer review of "Enhancing Motor and Sensory Axon Regeneration after Peripheral Nerve Injury Using Bioluminescent Optogenetics"

_ijms, 2022, doi:10.3390/ijms232416084_

Round 1

Reviewer 1 Report (Previous Reviewer 3)

Dear Authors,

It was a study on a very interesting topic. Although I think that sufficient evidence has not been presented and that additional experiments are necessary.

However, since the possibility was presented, I look forward to the follow experiment.

Thank you and best regards,

Author Response

It is a bit difficult to know exactly how to respond to this review, as no specifics are provided. However, we have made extensive changes to the text including:

  1. Re-writing of the Abstract in a structured format
  2. Addition of text to the Introduction to correlate better with Figure 1.
  3. Splitting the original figure 1 into two separate figures.
  4. Replacing Figure 3 (formerly Figure 2) with a higher quality image.
  5. Editing the Discussion to emphasize the major limitation of the study and to better summarize the conclusions drawn.
  6. Added references to the Introduction and Discussion sections.

Reviewer 2 Report (Previous Reviewer 2)

The article entitled “Enhancing Motor and Sensory Axon Regeneration after Peripheral Nerve Injury Using Bioluminescent Optoge-netics”. The aim of this review was to investigate whether similar results could be achieved using BL-OG when using a viral vector to induce luminopsin expression either before or after PNI.

Below are some suggestions:

In the Abstract:

- I suggest that the abstract be rewritten: it could be a structured and more organized abstract, as it is not clear the purpose of the research, as well as the methodology and the results. The relevance and clinical application of the research can also be inserted.

- Due to the importance of the repair process of peripheral nerve injuries, the authors could insert a paragraph emphasizing the importance of new techniques and therapeutic approaches already here in the abstract.

1. Introduction:

- The introduction is clear and objective, I suggest just improving the quality of the written part referring to figure 1 for better visualization.

2. Results

- 2.1 BL-OG Enhances Regeneration of Motor Axons: Improve the quality of figure 2 (graphics legend). I believe the paragraph (lines 61-65) is more for discussion.

- 2.4 Post-injury induction of BL-OG Promotes Recovery of Neuromuscular Function: I suggest staying more specifically in the results obtained because there are paragraphs that discuss ideas.

* I suggest, in general, separating the plates for better quality of images and graphics

3. Discussion

- The discussion is well written, comparing the data found in the research with data from the literature. I suggest just entering the search limitations. In conclusion, make a brief summary of what was investigated, ending with statements in accordance with the purpose of the research.

4. Materials and Methods

- The methodology is well described, I suggest inserting the number of the ethics committee's approval report.

*Regarding references, authors should increase the number of references and insert them in the text, since it is a journal with a high impact factor.

Author Response

Comments and Suggestions for Authors Responses are in RED

The article entitled “Enhancing Motor and Sensory Axon Regeneration after Peripheral Nerve Injury Using Bioluminescent Optoge-netics”. The aim of this review was to investigate whether similar results could be achieved using BL-OG when using a viral vector to induce luminopsin expression either before or after PNI.

Below are some suggestions:

In the Abstract:

I suggest that the abstract be rewritten: it could be a structured and more organized abstract, as it is not clear the purpose of the research, as well as the methodology and the results. The relevance and clinical application of the research can also be inserted.

- Due to the importance of the repair process of peripheral nerve injuries, the authors could insert a paragraph emphasizing the importance of new techniques and therapeutic approaches already here in the abstract.

The Abstract is re-written in a structured format to emphasize the purpose of the research, the methodology and the results, and the clinical significance of the findings. Text was added to the Introduction to emphasize bioluminescent optogenetics as a new approach with therapeutic potential.

  1. Introduction:

- The introduction is clear and objective, I suggest just improving the quality of the written part referring to figure 1 for better visualization.

  Text is added (lines 23-27, 29, 32-33, 35-36) to provide more details concerning the luminopsin diagram in the revised Figure 1.

  1. Results

- 2.1 BL-OG Enhances Regeneration of Motor Axons: Improve the quality of figure 2 (graphics legend). I believe the paragraph (lines 61-65) is more for discussion.

 The images in this figure have been replaced with higher quality (300 dpi) versions. Our original must not have been included at this resolution. Thank you for this suggestion. The sentence in question has been revised (new manuscript line 85)

- 2.4 Post-injury induction of BL-OG Promotes Recovery of Neuromuscular Function: I suggest staying more specifically in the results obtained because there are paragraphs that discuss ideas.

We believe that this paragraph simply describes the results obtained. We agree that ideas about these results are discussed elsewhere in the paper.

* I suggest, in general, separating the plates for better quality of images and graphics

 Previous Figure 1 is now split into two figures (new Figures 1 and 2), as they are used to make separate points in the text. We prefer to keep all other figures as multiple panels. Each is used to illustrate different sections of the Results section and separating them seems illogical. Old Figures 2-5 have been renumbered to Figures 3-6. Legends and text references have been altered accordingly. All figures have been checked to make sure that they include 300 dpi images.

  1. Discussion

- The discussion is well written, comparing the data found in the research with data from the literature. I suggest just entering the search limitations. In conclusion, make a brief summary of what was investigated, ending with statements in accordance with the purpose of the research.

 The text is modified in the Discussion section (lines 245-247, 251-253, 255) to emphasize the major limitation of the study: that we were unable to correlate the enhancement of axon regeneration produced by BL-OG with the extent of expression of eLMO3 or R115A in the same cells.

Text is added to the end of the Discussion section (lines 272-273, 275-276) to summarize our findings and relate them to the goal of the research.

  1. Materials and Methods

The methodology is well described, I suggest inserting the number of the ethics committee's approval report.

The number of the approved protocol from the Emory University Institutional Animal Care and Use Committee, the equivalent of an ethics committee for animal experiments, is already listed at the end of the manuscript (lines 404-405), as mandated in the Instructions to Authors.

 *Regarding references, authors should increase the number of references and insert them in the text, since it is a journal with a high impact factor.

Additional references are now added to the Introduction and Discussion sections.

Round 2

Reviewer 2 Report (Previous Reviewer 2)

Thank you very much for making all the changes that were suggested.

This manuscript is a resubmission of an earlier submission. The following is a list of the peer review reports and author responses from that submission.

Round 1

Reviewer 1 Report

Overall, this is an interesting paper that describes the possible action of spinal motoneurons expressing excitatory luminopsins on the regeneration of motor and sensory neurons after cut-inducing injury of the sciatic nerve. The methods are appropriate. The authors found that luminopsins enhanced the regeneration of the axons of motor and sensory neurons after peripheral nerve injury.

The results presented in the manuscript may influence the field, however, there are critical errors in the manuscript which lessen the potential impact this study will have on the field. 

With the idea to help to improve the impact of the findings reported here I suggest taking into consideration the next,

1.           In this research was used Bioluminescent optogenetics (BL-OG) to increase the activity of injured spinal motoneurons which is necessary and sufficient to improve axon regeneration. My main concern in this research is that even though it was found that BL-OG enhanced the regeneration of injured axons of motor and sensory neurons, it was not shown in this research that BL-OG increased the activity of spinal motoneurons and sensory neurons in the DRG, the lack of this evidence makes speculative the possible mechanism behind the enhanced in the motor-sensory regeneration neurons. Also, the low expression of luminopsins in motoneurons should be resolved before publishing at least as the authors suggest in the discussion section in line 159.

2.           Details in the methods section should be added. It was scored lumbar spinal motoneurons, but whether the whole lumbar enlargement was cut or only some length of the lumbar enlargement should be mentioned. Please specify whether the number of motoneurons in figure 2B is the average of how many slices.

3.           Please add which ANOVA was used for the analysis shown in Figures 2B and 3B. I assume 2WAY ANOVA because 2 variables were changing. 1) Axonal regeneration (intact vs injured axons) and 2) excitatory luminopsins (R115A vs eLMO3). If this is the case, please add whether was a significant interaction between the 2 variables. I think will help to the reader add the statistic used in the foot legend of the figures.

4.           Please add how was euthanized the mice.

5.           It was mentioned that the cohort includes 8 mice per group, however, in figures 2B, 2D, 3B, and 3C, not all groups have the same number of mice. What were the inclusion and exclusion criteria?

Author Response

  1. In this research was used Bioluminescent optogenetics (BL-OG) to increase the activity of injured spinal motoneurons which is necessary and sufficient to improve axon regeneration. My main concern in this research is that even though it was found that BL-OG enhanced the regeneration of injured axons of motor and sensory neurons, it was not shown in this research that BL-OG increased the activity of spinal motoneurons and sensory neurons in the DRG, the lack of this evidence makes speculative the possible mechanism behind the enhanced in the motor-sensory regeneration neurons. Also, the low expression of luminopsins in motoneurons should be resolved before publishing at least as the authors suggest in the discussion section in line 159.

We agree with the reviewer that our data interpretation is speculative as, in the current study, we did not specifically examine the physiological effect of BL-OG. However, the negative results with the control vector (point mutation R115A) eliminate most alternative scenarios and narrow down the possible mechanisms of BL-OG to a change in neuronal excitability because the only difference between the two conditions is cation-conduction that is possible with eLMO3 but not R115A. In fact, we had reported in a previous paper (English, et al., Int. J. Mol. Sci. 2021, 22, 7217-7234) that CTZ treatment of transgenic mice expressing the excitatory luminopsin, as well as mice in which eLMO3 expression was induced using viral vectors identical to those used in this study, produced an increase in the excitability of motoneurons, as measured by spinal evoked motor potentials, but did not increase their spontaneous activity. In the current study, we did not routinely evaluate the effectiveness of our CTZ treatments on the activity of spinal motoneurons and sensory neurons in the DRG. We assumed that the treatments with CTZ produced a similar increase in neuronal excitability as those observed in our previous study. Text has been added to the Methods section 4.1 (lines 192-193, 196-197) to make this point more clearly. Text has also been added to the conclusions section of the Discussion (line164) to indicate the need for future studies to resolve the issue of low expression of luminopsins in motoneurons.

  1. Details in the methods section should be added. It was scored lumbar spinal motoneurons, but whether the whole lumbar enlargement was cut or only some length of the lumbar enlargement should be mentioned. Please specify whether the number of motoneurons in figure 2B is the average of how many slices.

Text has been added to the Methods section 4.2 (lines 212 and 215) to indicate that the entire lumbar spinal cord was serially sectioned and that all sections were studied to count retrogradely labeled motoneurons.

  1. Please add which ANOVA was used for the analysis shown in Figures 2B and 3B. I assume 2WAY ANOVA because 2 variables were changing. 1) Axonal regeneration (intact vs injured axons) and 2) excitatory luminopsins (R115A vs eLMO3). If this is the case, please add whether was a significant interaction between the 2 variables. I think will help to the reader add the statistic used in the foot legend of the figures.

Individual one-way ANOVAs were run on the counts of labeled cells projecting to GAST and TA. Three groups were compared: Intact, eLMO3, and R115A. The following text was added to the legend for Figure 2: Significance of differences in means was evaluated using a one-way ANOVA, with post-hoc paired (Tukey’s) testing where appropriate.

  1. Please add how was euthanized the mice.

Text has been added to the Methods section 4.1 (lines 208-209) to indicate that euthanasia was induced by intraperitoneal injection of Euthasol (pentobarbital sodium and phenytoin sodium, 150 mg/Kg).

  1. It was mentioned that the cohort includes 8 mice per group, however, in figures 2B, 2D, 3B, and 3C, not all groups have the same number of mice. What were the inclusion and exclusion criteria?

Thank you for giving us the opportunity to correct errors in Figure 2B. In the experiments summarized in figure 2B, the sciatic nerve of one side of each mouse was cut and repaired and the contralateral sciatic nerve was left intact.  Four mice were injected with the eLMO3 vector and four mice were injected with the vector encoding R115A. This point is made in the revised manuscript (lines 172 and 173). Thus counts from intact mice are based on N=8, and for those from the two experimental groups, N=4. In the original Figure 2B, one count was accidently left out of the Intact group for TA and two mistaken zeros were included in the Intact group for the plot for neurons labeled with both tracers (Both). This has been corrected and Figure 2B has been revised. In Figure 2D, data are from the experimental (eLMO3 and R115A) groups only, where N=4. In Figure 3B, the same N=8 for Intact and N=4 for experimental as in Figure 3 are included. No data were excluded or included.

Reviewer 2 Report

The article entitled “Enhancing Motor and Sensory Axon Regeneration after Peripheral Nerve Injury Using Bioluminescent Optoge3 netics”. The aim of this study was to investigate whether similar results could be achieved using BL-OG when using a viral vector to induce luminopsin expression.

Below are some suggestions:

In the Abstract:

- I suggest rewriting the abstract with better organization of the ideas presented, as well as refining the methodology. The abstract should be more objective.

In the Introduction:

- The introduction is well written and objective, I suggest only emphasizing the importance and clinical applicability of the study.

In the Results:

- The results are well described as the quality of the images.

 In the Discussion:

- The discussion can be improved as it should compare the results of this research with data found in the literature. I also suggest entering the limitations of the study.

In the Materials and Methods:

- Do the authors consider n=8 sufficient?

- Do you have the number of the opinion issued by the ethics committee?

- Four weeks after nerve repair and CTZ treatment, different retrograde fluorescent markers were injected......can explain why after 4 weeks?

·        I suggest inserting a conclusion with final considerations, including the purpose of the research, its main results and future clinical perspectives.

Author Response

In the Abstract:

- I suggest rewriting the abstract with better organization of the ideas presented, as well as refining the methodology. The abstract should be more objective.

The abstract has been entirely rewritten with the hope that it is more objective.

In the Introduction:

- The introduction is well written and objective, I suggest only emphasizing the importance and clinical applicability of the study.

A sentence has been added to the end of this section (line 33) to emphasize this point.

In the Results:

- The results are well described as the quality of the images.

 In the Discussion:

- The discussion can be improved as it should compare the results of this research with data found in the literature. I also suggest entering the limitations of the study.

We thought that we had done a credible job of comparing our findings to those in the literature. We have added a sentence further emphasizing this point (lines 113-114). Similarly, we emphasize: 1) that we attribute the enhancement of regeneration to an increase in neuronal excitability induced using BL-OG (lines 134-144) and 2) that clinical translation of our findings will require induction of luminopsin expression after injury (lines 163-164) are both significant limitations of the significance of this study.

In the Materials and Methods:

- Do the authors consider n=8 sufficient?

The number of mice used was considered adequate, based on a power sample size estimate (power>0.8). A statement to this effect is now in the Methods section (lines 232-233).

- Do you have the number of the opinion issued by the ethics committee?

The number is given at line 257.

- Four weeks after nerve repair and CTZ treatment, different retrograde fluorescent markers were injected......can explain why after 4 weeks?

This is now explained in the revised Methods (lines 200-202).

         I suggest inserting a conclusion with final considerations, including the purpose of the research, its main results and future clinical perspectives.

The final paragraph of the Discussion (lines 160-166) has now been formatted as a conclusion.

Reviewer 3 Report

Major

1. I am curious about the behavior change before and after the procedure. It would be better if you attach the rotarod results.

2.  Please add data for nerve fibers in the surgical site. You focused on observing the cell body using virus tracing, but no results were obtained at the surgical site.

Minors.

1. It seems that the explanation of the terms GAST, TA, which suddenly appears on line 59, should be in the previous sentence.

2. In figure 3a, it may be helpful to understand not only the eLMO3 result, but also the r115A images. And, "GFP" is missing in green image.

3. In figure 3a, could you please present more high resolution image?

4. The number of experiments is too small. Please increase the number.

Author Response

  1. I am curious about the behavior change before and after the procedure. It would be better if you attach the rotarod results.

As it is not clear that rotarod experiments would constitute a sufficient assay for the efficacy of the BL-OG treatments on axon regeneration, they were not performed. However, in an earlier paper (English, et al., Int. J. Mol. Sci. 2021, 22, 7217-7234), we did show that a similar viral induction of eLMO3 and CTZ administration after sciatic nerve injury resulted in compound muscle action potentials that were significantly larger than controls. This paper is cited throughout the revised manuscript.

  1. Please add data for nerve fibers in the surgical site. You focused on observing the cell body using virus tracing, but no results were obtained at the surgical site.

We did not perform counts of axons at the surgical site in these experiments because a clear interpretation of results would be difficult or even impossible. An increase in these numbers produced by BL-OG could be due to: 1) increased branching of regenerating axons, as we have shown elsewhere; 2) the successful regeneration of axons of more neurons than controls; or 3) both. We focused on retrograde tracing because the results provide unambiguous evidence of the extent of successful axon regeneration.

 Minors.

  1. It seems that the explanation of the terms GAST, TA, which suddenly appears on line 59, should be in the previous sentence.

Definitions of GAST and TA are now made prior to their first usage (lines 35- 38). Thanks for finding this oversight.

  1. In figure 3a, it may be helpful to understand not only the eLMO3 result, but also the r115A images. And, "GFP" is missing in green image.

The “GFP” has been recovered in this figure.

  1. In figure 3a, could you please present more high resolution image?

The images in revised Figure 3A have been formatted to higher resolution.

  1. The number of experiments is too small. Please increase the number.

 We respectfully disagree with his concern. A power sample size estimate was run on the collected data and resulted in a power of >0.8 at p<0.05. A statement to this effect is now in the Methods section (lines 232-233).

Round 2

Reviewer 1 Report

The authors responded to all my concerns.